**Subject Category:**
Biology (whole organism)

behaviour/ecology/evolution

agonistic behaviour, territoriality, heterospecific aggression

**Author for correspondence:**
Brent L. Lockwood
e-mail: brent.lockwood@uvm.edu

# Aggression and discrimination among closely versus distantly related species of *Drosophila*

Tarun Gupta, Sarah E. Howe, Marlo L. Zorman and Brent L. Lockwood

Department of Biology, University of Vermont, 109 Carrigan Dr., Burlington, VT 05405, USA

 BLL, 0000-0003-4694-5897

Fighting between different species is widespread in the animal kingdom, yet this phenomenon has been relatively understudied in the field of aggression research. Particularly lacking are studies that test the effect of genetic distance, or relatedness, on aggressive behaviour between species. Here we characterized male–male aggression within and between species of fruit flies across the *Drosophila* phylogeny. We show that male *Drosophila* discriminate between conspecifics and heterospecifics and show a bias for the target of aggression that depends on the genetic relatedness of opponent males. Specifically, males of closely related species treated conspecifics and heterospecifics equally, whereas males of distantly related species were overwhelmingly aggressive towards conspecifics. To our knowledge, this is the first study to quantify aggression between *Drosophila* species and to establish a behavioural bias for aggression against conspecifics versus heterospecifics. Our results suggest that future study of heterospecific aggression behaviour in *Drosophila* is warranted to investigate the degree to which these trends in aggression among species extend to broader behavioural, ecological and evolutionary contexts.

## 1. Introduction

Heterospecific aggression—i.e. fighting between members of different species—is widespread in the animal kingdom [1]. Aggressive behaviour often mediates competitive interactions between species that can have important consequences for species coexistence and the structure of ecological communities [2–5]. Yet, most research into aggressive behaviour has focused on conspecific aggression—i.e. fighting between members of the same species [6]—with few well-characterized examples of heterospecific aggression [7], particularly in a broad phylogenetic context [8].

**Figure 1.** Phylogenetic relationships among focal species. Branch lengths reflect the number of substitutions per nucleotide site, as indicated by the scale bar.

Fruit flies in the genus *Drosophila* present a unique opportunity to investigate aggressive behaviours, both within and between species and in a broad phylogenetic context. There are approximately 1500 described species of *Drosophila*, many of which overlap spatially and temporally [9–11] and use similar territories and food resources for feeding, breeding and ovipositing [12]. Yet, while it is well established that *Drosophila* use aggression within species to establish territories and social dominance and to compete for mates and food resources [13–18], heterospecific aggression is largely uncharacterized, except for limited qualitative observations of heterospecific aggression among the Hawaiian *Drosophila* [19].

Here we characterized male–male aggression in *Drosophila* in a multi-species context using a behavioural choice assay, in order to (i) quantify the extent to which male *Drosophila* discriminate between conspecifics and heterospecifics during aggressive social interactions and (ii) test the effect of phylogenetic distance between opponent species on the distributional bias in aggressive targeting (heterospecific versus conspecific). We report that males showed significant bias in aggression towards either conspecifics or heterospecifics in a majority of species–species interactions. Among species pairs that were more distantly related, the direction of aggression was biased towards conspecifics, whereas closely related species treated conspecifics and heterospecifics equally. To our knowledge, this is the first study to quantify aggression between *Drosophila* species and to establish a behavioural bias for aggression against conspecific versus heterospecific opponents.

# 2. Material and methods

## 2.1. *Drosophila* species and husbandry

Seven species were selected from the *ananassae*, *melanogaster* and *pseudoobscura* subgroups within the subgenus *Sophophora* (figure 1). Among these seven species, we assayed aggressive interactions between two species at a time for a total of six species pairs. Three of these species pairs are relatively closely related sibling species: (i) *D. ananassae* and *D. pallidosa*, (ii) *D. melanogaster* and *D. simulans*, and (iii) *D. pseudoobscura* and *D. persimilis*. Whereas the other three species pairs are more distantly related: (i) *D. ananassae* and *D. atripex*, (ii) *D. ananassae* and *D. melanogaster*, and (iii) *D. ananassae* and *D. simulans* (figure 1). All seven species have broad geographical distributions, and for each species pair the geographical distributions overlap [20]. To the best of our knowledge, none of these species exhibit lekking behaviour. Male–male aggression has previously been documented in *D. melanogaster* and *D. simulans* [15,21], but not in the other five species. We used one isogenic (isofemale) line from each species that was originally established from wild collections. *Drosophila melanogaster* was obtained from the Bloomington Stock Center (Bloomington, IN; Canton-S; Stock No.: 64349). The *D. simulans* stock was generously provided by Brandon Cooper and Michael Turelli. The following stocks were obtained from the *Drosophila* Species Stock Center (San Diego, CA, USA): *D. ananassae* (14024-0371.15), *D. pallidosa* (14024-0433.00), *D. atripex* (14024-0361.03), *D. pseudoobscura* (14011-0121.148) and *D. persimilis* (14011-0111.50). All flies were reared on cornmeal-based fly food containing: 1% (w/v)

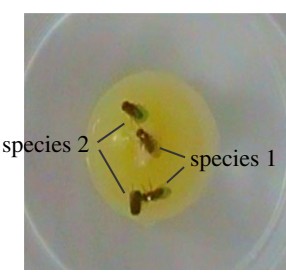
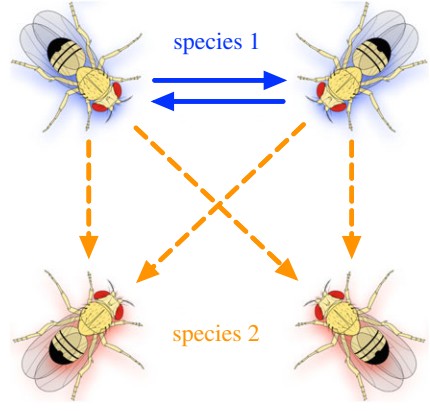

**Figure 2.** Multi-species aggression assay. Left panel is an image of the aggression arena showing separate species labelled on the thoraxes by white paint (visible in the image, species 1) or blue paint (not visible in the image, species 2). Right panel illustrates the null expectation of a 2 : 1 ratio of the number of heterospecific lunges to conspecific lunges. Note that in the statistical analysis the number of heterospecific lunges was normalized by dividing by 2 to account for this 2 : 1 ratio of encounters among males.

agarose, 8% (w/v) yeast, 4.5% molasses and 10% (w/v) cornmeal in standard $25 \times 95$ mm polystyrene vials under humidity- and temperature-controlled conditions (25°C, 50% humidity and 12 : 12 h light–dark cycles). *Drosophila pseudoobscura* and *D. persimilis* were reared at 19°C, 50% humidity and 12 : 12 h light–dark cycles.

## 2.2. Aggression assay

To quantify agonistic social interactions in a multi-species context, we used a slightly modified version of the standard dyadic aggression assay [22]. For each species pair, aggressive behaviours were quantified by placing two socially naive adult males from each opponent species—a total of four males—in a standard aggression arena (figure 2) and measuring (i) the delay to onset of aggression (latency to aggression) and (ii) the total number of aggressive lunges—a key indicator of aggression [21]—by each male towards both conspecific and heterospecific opponents. We examined a total of twelve sets of interactions, as we tracked aggressive behaviours for each focal species across six species pairs. In contrast to dyadic assays typically employed in aggression studies in *Drosophila* [23,24], our multi-individual, multi-species paradigm allows examination of social behaviour in a context where multiple individuals from different species compete for shared resources or territory, and it also allows us to quantify choice behaviour—i.e. bias in aggression towards heterospecifics versus conspecifics. Male pupae were isolated in $16 \times 100$ mm borosilicate glass tubes containing 1.5 ml of standard food medium and aged individually for 3–4 days to prevent social conditioning or formation of social dominance hierarchies prior to testing. Three-day-old adult males were extracted under $CO_2$ anaesthesia and marked on the thorax with a dab of white or blue acrylic paint (assigned randomly) for species identification during assay set-up and scoring. After painting, males were transferred to new isolation tubes containing 1.5 ml agarose-based nutritionally deficient media (without cornmeal, yeast or sugar) and allowed recovery from handling and anaesthesia. The following day, two 4–5-day-old, socially naive adult males from each opponent group—a total of four males—were gently aspirated into one of the wells of a 12-well polystyrene plate (Thermo Fisher Scientific #130185) with a small cup in the middle containing food—representing focal point of contest (figure 2). All four males were introduced to the chamber at the same time to prevent a potential resident–intruder confound. All behavioural assays were set up and recorded within 0–2 Zeitgeber hours, i.e. the first two hours of the lights ON time in a 12:12 light–dark cycle.

## 2.3. Aggression scoring

The number of lunges against conspecific and heterospecific males was counted for a period of 30 min after the first lunge, for consistency with the scoring duration of aggression assays reported elsewhere [23,24]. The amount of time between the introduction of males to the aggression chamber and the first aggressive lunge was used as the measurement of delay to the onset of aggression, or latency to lunge. The latency to lunge was scored separately for the two directions of lunges, conspecific or

heterospecific. Scoring was terminated after 1 h if no aggressive encounter was recorded during that period. Electronic supplementary material, figure S3 shows the proportion of aggression trials in which lunges were recorded. Aggressive behaviours were scored manually by two independent scorers using iMovie '09, version 8.0.6 (Apple Inc., Cupertino, CA, USA). The number of aggression trials is indicated in electronic supplementary material, figures S1 and S2.

## 2.4. Phylogenetic distance

We inferred evolutionary relationships (figure 1) and calculated pairwise genetic distances (figure 5) among species using the maximum-likelihood method based on the Tamura–Nei model [25] using MEGA7 [26]. We compared 3196 nucleotide positions among one mitochondrial gene (*CoI*) and two nuclear genes (*Gpdh* and *kl2*), for which sequence data were readily available on NCBI for all species included in the study. Although full genome sequences are available for *D. melanogaster*, *D. simulans*, *D. pseudoobscura*, *D. persimilis* and *D. ananassae*, there is relatively little nucleotide sequence data available for *D. pallidosa* and *D. atripex*. Nonetheless, the evolutionary relationships that we report herein are consistent with previous studies [27,28]. We downloaded sequences from NCBI of *D. pallidosa* (*CoI*, Accession no.: FJ795561; *Gpdh*, FJ795596; and *kl2*, FJ795633) and *D. atripex* (*CoI*, FJ795575; *Gpdh*, FJ795601; and *kl2*, FJ795643) and used these to BLAST the published genome sequences of the other five species included in the study. The phylogenetic tree with the highest log likelihood (−6435.05) is shown in figure 1. Initial trees for the heuristic search were obtained by Neighbor-Join and BioNJ to a matrix of pairwise distances estimated with the maximum composite likelihood approach. A discrete Gamma distribution was used to model evolutionary rate differences among sites. All positions containing gaps were removed from the analysis. Branch lengths correspond to the number of substitutions per site.

## 2.5. Body-size estimation

In many species, aggressiveness correlates with body size both within and between species and smaller males are less likely to initiate and hold aggressive encounters [15]. Body length (mm) from anterior antennae to posterior abdomen of males from all opponent groups used in aggression assays was measured as a proxy for body size using ImageJ [29].

## 2.6. Statistical analyses

We compared the number of lunges directed towards conspecifics versus heterospecifics with a negative binomial generalized linear model, as implemented in the MASS package in R version 3.3.2 (R Core Team 2016 [30]). We normalized the number of heterospecific lunges by dividing by two (rounded to the nearest whole number) because there were twice as many heterospecific opponent males as conspecific opponent males in the aggression arena (figure 2). Goodness-of-fit was assessed by a chi-square test of the residual deviance of the negative binomial model. To examine the direction and effect size of aggression bias, we calculated the ratio of mean lunge counts (*RL*), which is the ratio of the mean number of heterospecific lunges to the mean number of conspecific lunges. *RL* was obtained by exponentiating the regression coefficient ($\beta$) of the negative binomial model, as this coefficient equals the log-ratio of mean lunge counts. The negative binomial model was also used to estimate the 95% confidence intervals of *RL*, which are reported in figure 3. Significant differences between the distributions of heterospecific and conspecific lunge counts were determined by a *z*-test of the estimated regression coefficient and standard error from the negative binomial model. We ran separate regressions for each focal species in each species pair for a total of 12 regressions (6 species pairs × 2 species in each pair). *p*-Values were corrected for multiple tests via the Benjamini–Hochberg method [31].

We assessed the relationship between aggression bias (*RL*) and genetic distance between opponent species via permutation analyses. Given the design of our aggression assay and the phylogenetic relatedness of the focal species, aggression biases among species pairs may not be independent. To account for this lack of independence among data points in our analyses, we performed 10 000 permutations of the genetic distance versus *RL* relationships—i.e. the genetic distances and *RL* values for each species pair were randomly shuffled and resampled—and significance was assessed based on the probability distribution of the Spearman rank coefficient.

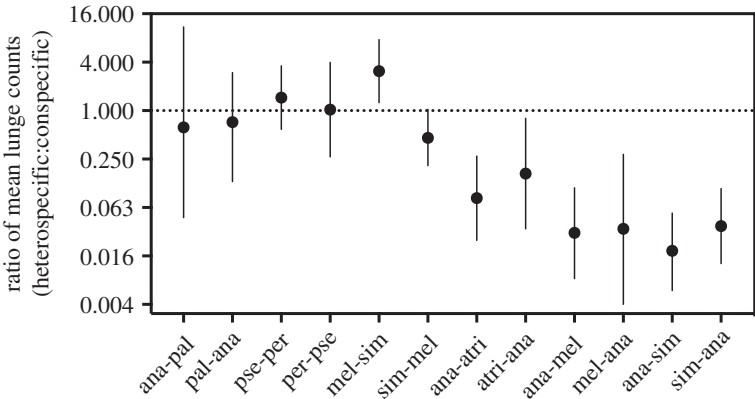

**Figure 3.** Ratio of mean lunge counts for each species pair interaction. The dotted line at 1 indicates the null expectation that heterospecifics and conspecifics were treated equally. Values greater than 1 indicate greater aggression towards heterospecifics, and values less than 1 indicate greater aggression towards conspecifics. The data are presented on a log scale. Focal species is the aggressor that lunges on itself or the non-focal species (focal-nonfocal), with the same species name abbreviations as in table 1. Species pairs are ordered from most closely related to most distantly related (left to right). Error bars represent 95% confidence intervals.

We examined the relationship between the latency to lunge and the direction of the first lunge (conspecific versus heterospecific) via a two-way ANOVA, with species pair and direction of lunge as fixed effects. We examined the relationships between (i) body length and number of lunges from focal males and (ii) body length difference between opponent species and total number of heterospecific lunges via the Spearman correlation. All analyses were conducted in R version 3.3.2 (R Core Team 2016) or GraphPad Prism version 7 (GraphPad Software, Inc., La Jolla, CA, USA).

# 3. Results

We observed a significant distributional bias in the targets of aggression—i.e. lunges directed towards either conspecific or heterospecific opponent males—in seven out of twelve species-pair interactions (table 1). The behaviour of closely related species pairs contrasted with that of more distantly related species pairs. Among closely related species pairs, heterospecifics and conspecifics were treated more or less equally (i.e. there was not a strong bias in the direction of aggression), as can be seen in the largely overlapping distributions of heterospecific and conspecific lunge counts (electronic supplementary material, figure S1). In addition, the ratios of mean lunge counts ($RL$; heterospecific : conspecific) in closely related species pairs hovered around values of one (figure 3; $RL \approx 1$), indicating that heterospecifics and conspecifics were equally likely to be targeted by aggression. The only closely related species pair interaction that showed a significant aggression bias was *D. melanogaster* paired with *D. simulans* (table 1), where *D. melanogaster* males were three times more likely to target heterospecifics than conspecifics (figure 3 and electronic supplementary material, figure S1; $RL = 3.09$). In contrast, among more distantly related species pairs, the distributions of heterospecific and conspecific lunges did not overlap (electronic supplementary material, figure S2), and the ratios of mean lunge counts were all less than one (figure 3; $RL < 1$), indicating strong conspecific aggression biases. These patterns of conspecific aggression bias were also reflected by the number of lunges per aggression trial (figure 4). In addition, species that were included in multiple species-pair interactions (i.e. *D. ananassae*, *D. melanogaster* and *D. simulans*) were not more or less aggressive overall than other species (figure 4). Rather, for species included in multiple species pair interactions, the level and direction of aggression depended on the opponent species (table 1, figures 3 and 4).

Further supporting the contrast between the aggressive behaviours of closely versus distantly related species pairs, there was a significant negative relationship between the genetic distance between competing species and the ratio of mean lunge counts (figure 5; Spearman $\rho = -0.82$, $N = 10\,000$ permutations, $p = 0.002$). In other words, more distantly related species pairs were most aggressive to conspecifics, whereas closely related species pairs treated conspecifics and heterospecifics with equal levels of aggression. In fact, males in the distantly related *D. simulans*–*D. ananassae* species pair displayed a high degree of tolerance for heterospecific opponents sharing the food cup but escalated quickly to high-intensity lunging when confronted by conspecific opponents (electronic supplementary

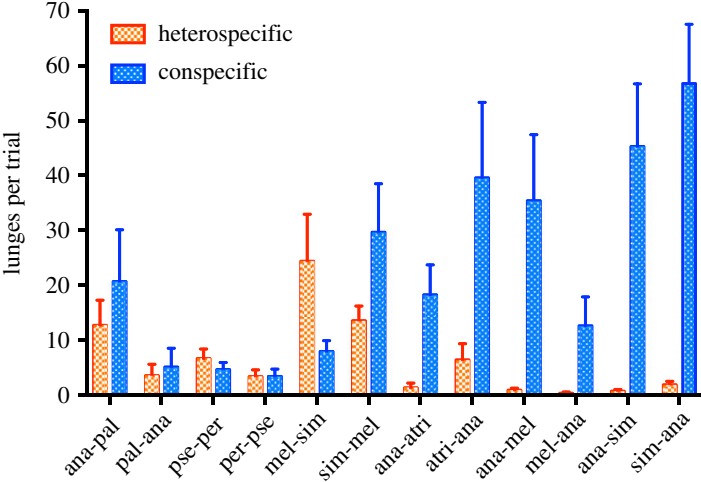

**Figure 4.** Mean number of lunges per aggression trial. Focal species is the aggressor that lunges on itself or the non-focal species (focal-nonfocal), with the same species name abbreviations as in table 1. Species pairs are ordered from most closely related to most distantly related (left to right). Error bars represent standard error of the mean.

**Table 1.** Statistics of the negative binomial model fits. Values in italics indicate significant differences between the distributions of heterospecific and conspecific lunge counts after false discovery rate correction. Focal species is the aggressor that lunges on itself or the non-focal species (focal-nonfocal), with the following species name abbreviations: mel = *D. melanogaster*, sim = *D. simulans*, pse = *D. pseudoobscura*, per = *D. persimilis*, pal = *D. pallidosa*, ana = *D. ananassae*, atri = *D. atripex*. Species pairs are ordered from most closely related to most distantly related (top to bottom).

| focal-nonfocal | residual deviance | residual d.f. | goodness of fit ($X^2$ $p$-value) | $\beta \pm$ std. error | $z$ | FDR-corrected $p$-value |
| --- | --- | --- | --- | --- | --- | --- |
| ana-pal | 27.40 | 24 | 0.2860 | $-0.47 \pm 0.76$ | $-0.62$ | 0.6443 |
| pal-ana | 16.40 | 24 | 0.8732 | $-0.33 \pm 1.24$ | $-0.27$ | 0.8602 |
| pse-per | 58.33 | 54 | 0.3193 | $0.37 \pm 0.46$ | $0.82$ | 0.5484 |
| per-pse | 43.17 | 54 | 0.8545 | $0.03 \pm 0.67$ | $0.05$ | 0.9633 |
| mel-sim | 63.20 | 54 | 0.1833 | *$1.13 \pm 0.45$* | *2.49* | *0.0255* |
| sim-mel | 65.46 | 54 | 0.1364 | $-0.77 \pm 0.41$ | $-1.89$ | 0.0890 |
| ana-atri | 38.66 | 38 | 0.4396 | *$-2.50 \pm 0.60$* | *$-4.15$* | *0.0001* |
| atri-ana | 37.25 | 38 | 0.5041 | *$-1.80 \pm 0.77$* | *$-2.33$* | *0.0336* |
| ana-mel | 28.43 | 26 | 0.3378 | *$-3.50 \pm 0.65$* | *$-5.39$* | *$2.86 \times 10^{-7}$* |
| mel-ana | 18.76 | 26 | 0.8466 | *$-3.38 \pm 1.04$* | *$-3.26$* | *0.0027* |
| ana-sim | 22.82 | 20 | 0.2977 | *$-4.01 \pm 0.56$* | *$-7.18$* | *$8.64 \times 10^{-12}$* |
| sim-ana | 25.20 | 20 | 0.1940 | *$-3.30 \pm 0.54$* | *$-6.12$* | *$5.53 \times 10^{-9}$* |

material, video S1). Conversely, the intensity of aggression directed towards heterospecifics was greatest in closely related species pairs, such as *D. simulans–D. melanogaster* (electronic supplementary material, video S2).

There was no significant difference in the latency to initiate aggression towards conspecifics or heterospecifics (electronic supplementary material, figure S4; two-way ANOVA, direction of lunge effect, $F_{1,177} = 0.00057$, $p = 0.98$, direction of lunge × species pair interaction, $F_{5,177} = 1.515$, $p = 0.19$). That is, males from either opponent group were equally likely to be targeted at the initial onset of aggression when low-intensity encounters first escalate to high-intensity lunging. These results suggest an opportunistic, non-selective tendency towards initiating an aggression sequence followed by a species-specific strategy for selectively targeting subsequent aggressive behaviours.

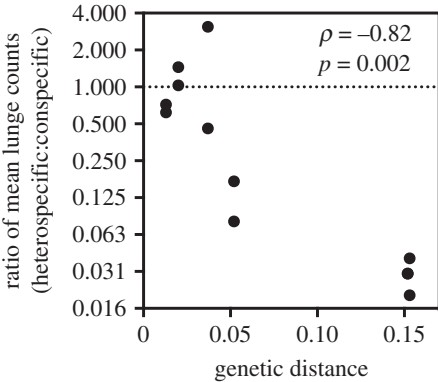

**Figure 5.** Greater genetic distance between species pairs led to significant conspecific aggression biases (Spearman $\rho = -0.82$, $N = 10\,000$ permutations, $p = 0.002$). Plotted are the pairwise genetic distances versus the mean ratio of lunge counts for each species pair. The dotted line at 1 indicates the null expectation that heterospecifics and conspecifics were treated equally. Values greater than 1 indicate greater aggression towards heterospecifics, and values less than 1 indicate greater aggression towards conspecifics.

Body size differences among opponents have been shown previously to influence male fly aggressiveness [15], but we found no significant association between average body size and number of aggressive lunges by a given species (electronic supplementary material, figure S5B; Spearman $\rho = 0.015$, $N = 155$, $p = 0.85$). Furthermore, the relative body-size difference between opponent species in a given fight showed no significant relationship to the number of heterospecific lunges (electronic supplementary material, figure S5C; Spearman $\rho = -0.14$, $N = 75$, $p = 0.24$).

## 4. Discussion

To our knowledge, this is the first study to demonstrate discriminatory aggression between species of *Drosophila*—i.e. the differential aggressive response of males towards conspecifics versus heterospecifics in multi-species social interactions—albeit aggression biases were mostly observed between distantly related species and not closely related species. While males of many species of *Drosophila* are known to be territorial [15,17,24], particularly in the lekking species that are endemic to Hawaii [32], previous work has only provided limited accounts of heterospecific interactions [19], and heterospecific aggression has never been explicitly quantified.

We interpret the differential aggressive responses among closely versus distantly related species pairs as innate responses that are mediated by species recognition cues. Because all interacting individuals in this study were extracted as pupae and socially isolated as adults with no direct contact with other males from either species, the biases in aggressive targeting are not likely to be learned behaviours. Several potential molecular mechanisms may underlie these behavioural responses to male–male encounters of different species. Recently, it has been reported that epigenetic mechanisms, such as DNA methylation, serve as an interface between the genome and the environment and can facilitate species-specific behavioural plasticity in the context of courtship by modulating aminergic function [33]. Thus, as a proximate mechanistic cause for the bias in aggressive targeting reported in the present study, octopaminergic systems may play a critical role in relaying species-specific chemosensory information [34], and facilitate species recognition and/or discrimination in the context of mixed-species aggressive interactions. Furthermore, the cues that stimulate the neural substrates of species recognition and subsequent aggressive targeting may incorporate pheromone cues, a mechanism that has been previously shown to mediate aggression among *D. melanogaster* conspecifics [35]. In other words, males of closely related species may treat each other as conspecifics simply because they smell alike, a case of mistaken identity (see below).

Based solely on the data presented herein, we cannot evaluate the ultimate evolutionary causes of male aggression biases among *Drosophila* spp. Nonetheless, it is important to consider the potential ecological and evolutionary mechanisms that influence these patterns in order to provide a framework for future work. A major outstanding question is whether these behavioural biases for aggression are due to ancestral states, where males treat closely related heterospecifics like conspecifics due to mistaken identity (i.e. falsely identifying a heterospecific opponent as a conspecific), or if aggression

bias is influenced by current and ongoing interference competition. Previous studies in other animal species lend support to the mistaken identity hypothesis. In a meta-analysis of birds and fish, Peiman & Robinson [1] found that, among species that do not share resources, heterospecific aggression is greatest among closely related species. Similarly, in a separate meta-analysis of wood warbler birds, Losin et al. [36] found that, even among sympatric species, patterns of heterospecific aggression can largely be explained by shared ancestry. Thus, in many cases, heterospecific aggression may be an evolutionary artefact that originates from natural selection for conspecific aggression, which erodes over time following species divergence. In fact, it may be difficult to parse this non-adaptive cause of heterospecific aggression from the effects of interference competition between species. To overcome this challenge and potentially account for these confounding effects, Peiman & Robinson [1] suggest comparing levels of heterospecific aggression among allopatric species versus aggression among sympatric species. In the allopatric case, heterospecific aggression should be non-adaptive because species do not directly compete for resources. In comparison to allopatry, higher or lower levels of heterospecific aggression in sympatry could be attributed to the evolution of aggressive behaviours in response to interference competition among species.

A further consideration in analyses of heterospecific aggression, and an important caveat to the interpretation of the results we report herein, is the degree to which species pairs directly interact in nature. All of the species pairs included in this study have large biogeographic ranges that overlap to varying degrees [20], and each species pair has similar food preferences [9,10,12]. Thus, it is possible that these species pairs compete in nature, and that direct interference competition between species influences the evolution of heterospecific aggression. However, to our knowledge direct interference competition among the six species pairs included in this study has never been documented in nature. Therefore, future work is required to address whether or not these species pairs directly compete for resources in nature. If, in fact, these species directly compete in nature, then it is interesting to note that our results follow the general predictions of the limiting similarity hypothesis [3], which states that the greatest degree of interference competition exists between closely related species. Further, these trends in male–male aggression behaviour predict that more distantly related species would coexist without posing significant costs in energy and time devoted to heterospecific aggression.

We would also like to note that direct competition for mates (i.e. reproductive interference competition) among heterospecifics is another type of interference competition that could influence heterospecific aggression. Indeed, the intensity of reproductive competition among species has been shown to be a key factor that influences heterospecific aggression in other species [37]. We were not able to assess this effect in the present study because, while hybridization is known to occur in the laboratory among the closely related sibling species pairs [11], there is lack of consensus as to whether or not hybridization occurs in nature for these species pairs [11,27,38,39]. On a related note, in our aggression trials we intermittently observed what looked like courting behaviour between males of different species—e.g. D. simulans males exhibited courtship displays to D. melanogaster males that consisted of single wing extensions (electronic supplementary material, video S2). We did not score this behaviour because it is difficult to interpret the significance of these interactions in the context of male–male aggression. We did not observe escalation of this courtship-like behaviour to the point of attempted mounting; thus, it could be that the single wing extensions constituted a form of aggressive interaction and not courtship. If these single-wing extensions were in fact misdirected courtship towards heterospecific males, this behaviour could further complicate heterospecific interactions and even induce reproductive interference if this behaviour occurs in nature. To address these questions of potential reproductive interference competition, future work should focus on comparisons of taxa among sibling species of Drosophila that are known to hybridize in nature, such as species in the subquinaria complex [40] or among the yakuba, teissieri and santomea sister species [41,42], to examine the role of reproductive interference competition in influencing the evolution of heterospecific aggression.

Data accessibility. Raw data from the scoring of aggression videos and representative video clips of aggression trials are included as electronic supplementary material.

Authors' contributions. T.G. and B.L.L. conceived and planned the study. T.G., S.E.H. and M.L.Z. conducted the experiments. T.G. and B.L.L. analysed the data. T.G. and B.L.L. wrote the manuscript.

Competing interests. The authors declare no competing interests.

Funding. This work was supported by the University of Vermont.

Acknowledgements. We thank Brandon Cooper and Michael Turelli for generously providing the D. simulans stock. We thank Sara Helms Cahan, Nicholas Gotelli, Melissa Pespeni, Emily Mikucki, Dean Castillo, Leonie Moyle and six anonymous reviewers for their helpful discussions and comments on this manuscript.

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
