## [Reviewer comments · Royal Society Open Science]

Review History

RSOS-190069.R0 (Original submission)

Review form: Reviewer 1

Is the manuscript scientifically sound in its present form?

Yes

Are the interpretations and conclusions justified by the results?

Yes

Is the language acceptable?

Yes

Is it clear how to access all supporting data?

Yes

Do you have any ethical concerns with this paper?

No

Have you any concerns about statistical analyses in this paper?

No

Recommendation?

Accept with minor revision (please list in comments)

Comments to the Author(s)

1) A major concern in the previous round of review was that other possible combinations of "distantly related" species, such as pse-ana, pal-mel and pse-mel, were not examined. Although the authors answered as "we view this as an important next step in future work, as we state in the Discussion (Lines 303-325 and 349-353)", I don't think these parts are not directly related to this concern. I appreciate the new Fig. 4, but I can't find any statements referring to it in the main text. Why not include the following statements not only in the response to review but also into the manuscript?

"As can be seen in this new figure Fig. 4, *D. ananassae* was not more or less aggressive overall than other species, such as *D. melanogaster*, *D. simulans*, and *D. atriplex*. Furthermore, the level and direction of aggression (heterospecific vs. conspecific) depended on the opponent species – e.g., the level of conspecific aggression was higher when species were paired with more distantly related species for *D. ananassae*, *D. melanogaster*, and *D. simulans*. Thus, it does not appear that the species-pair interactions that include *D. ananassae* are what solely drive the main patterns of our results. It will be important in future work to investigate broader sets of species, and in all combinations of species-pair interactions, in order to evaluate these trends in aggression behavior more broadly. It will be important in future work to investigate broader sets of species, and in all combinations of species-pair interactions, in order to evaluate these trends in aggression behavior more broadly."

2) I enjoyed the supplemental videos, and noticed that *D. sim* males were rigorously courting *D. mel* males in the Video S2. I suspect it induced more heterospecific aggression from *D. mel* males towards *D. sim* males. Because it suggests a possibility that non-aggressive behavior affects the results, I hope to see some explanations in the main text. "Mis-directed courtship toward heterospecific males" could be a kind of reproductive interference if it happens under the natural condition.

Review form: Reviewer 2**Is the manuscript scientifically sound in its present form?**

Yes

Are the interpretations and conclusions justified by the results?

Yes

Is the language acceptable?

Yes

Is it clear how to access all supporting data?

No

Do you have any ethical concerns with this paper?

No

Have you any concerns about statistical analyses in this paper?

No

Recommendation?

Accept as is

Comments to the Author(s)

All my concerns have been addressed. The paper is clearer, and the figures are much improved. This study is a valuable contribution to our understanding of heterospecific aggression. I do think it's curious that there seems to be some interactive or condition-dependent effect of the species pair and conspecific aggression. I'm not sure what can be said about this, but it's interesting.

Decision letter (RSOS-190069.R0)

16-Apr-2019

Dear Dr Lockwood

On behalf of the Editors, I am pleased to inform you that your Manuscript RSOS-190069 entitled "Aggression and discrimination among closely versus distantly related species of *Drosophila*" has been accepted for publication in Royal Society Open Science subject to minor revision in accordance with the referee suggestions. Please find the referees' comments at the end of this email.

The reviewers and handling editors have recommended publication, but also suggest some minor revisions to your manuscript. Therefore, I invite you to respond to the comments and revise your manuscript.

- Ethics statement

- Data accessibility

<http://datadryad.org/submit?journalID=RSOS&manu=RSOS-190069>

- **Competing interests**

- **Authors' contributions**

- **Acknowledgements**

- **Funding statement**

Because the schedule for publication is very tight, it is a condition of publication that you submit the revised version of your manuscript before 25-Apr-2019. Please note that the revision deadline will expire at 00.00am on this date. If you do not think you will be able to meet this date please let me know immediately.

1) Identifying all the changes that have been made (for instance, in coloured highlight, in bold text, or tracked changes);

on behalf of Dr Ryan Earley (Associate Editor) and Professor Kevin Padian (Subject Editor)
openscience@royalsociety.org

Reviewer comments to Author:

Reviewer: 1

Comments to the Author(s)

1) A major concern in the previous round of review was that other possible combinations of "distantly related" species, such as *pse-ana*, *pal-mel* and *pse-mel*, were not examined. Although the authors answered as "we view this as an important next step in future work, as we state in the Discussion (Lines 303-325 and 349-353)", I don't think these parts are not directly related to this concern. I appreciate the new Fig. 4, but I can't find any statements referring to it in the main text. Why not include the following statements not only in the response to review but also into the manuscript?

"As can be seen in this new figure Fig. 4, *D. ananassae* was not more or less aggressive overall than other species, such as *D. melanogaster*, *D. simulans*, and *D. atripex*. Furthermore, the level and direction of aggression (heterospecific vs. conspecific) depended on the opponent species – e.g., the level of conspecific aggression was higher when species were paired with more distantly related species for *D. ananassae*, *D. melanogaster*, and *D. simulans*. Thus, it does not appear that the species-pair interactions that include *D. ananassae* are what solely drive the main patterns of our results. It will be important in future work to investigate broader sets of species, and in all combinations of species-pair interactions, in order to evaluate these trends in aggression behavior more broadly. It will be important in future work to investigate broader sets of species, and in all combinations of species-pair interactions, in order to evaluate these trends in aggression behavior more broadly."

2) I enjoyed the supplemental videos, and noticed that *D. sim* males were rigorously courting *D. mel* males in the Video S2. I suspect it induced more heterospecific aggression from *D. mel* males towards *D. sim* males. Because it suggests a possibility that non-aggressive behavior affects the results, I hope to see some explanations in the main text. "Mis-directed courtship toward heterospecific males" could be a kind of reproductive interference if it happens under the natural condition.

Reviewer: 2

Comments to the Author(s)

All my concerns have been addressed. The paper is clearer, and the figures are much improved. This study is a valuable contribution to our understanding of heterospecific aggression. I do think it's curious that there seems to be some interactive or condition-dependent effect of the species pair and conspecific aggression. I'm not sure what can be said about this, but it's interesting.

Author's Response to Decision Letter for (RSOS-190069.R0)

See Appendix A.

Decision letter (RSOS-190069.R1)

14-May-2019

Dear Dr Lockwood,

I am pleased to inform you that your manuscript entitled "Aggression and discrimination among closely versus distantly related species of *Drosophila*" is now accepted for publication in Royal Society Open Science.

on behalf of Dr Ryan Earley (Associate Editor) and Kevin Padian (Subject Editor)
openscience@royalsociety.org

Associate Editor Comments to Author (Dr Ryan Earley):
Associate Editor: 1
Comments to the Author:
(There are no comments.)

Reviewer comments to Author:

Appendix A

Gupta et al., *Aggression and discrimination among closely versus distantly related species of Drosophila*, RSOS-190069

Response to Review:

We have addressed all of the comments of the two anonymous reviewers in the revision of our manuscript. Below are the reviewer's comments (in italics), followed by our responses (in blue font). In the revised manuscript, all new text is highlighted in blue font.

Reviewer comments to Author:

Reviewer: 1

Comments to the Author(s)

1) A major concern in the previous round of review was that other possible combinations of "distantly related" species, such as pse-ana, pal-mel and pse-mel, were not examined. Although the authors answered as "we view this as an important next step in future work, as we state in the Discussion (Lines 303-325 and 349-353)", I don't think these parts are not directly related to this concern. I appreciate the new Fig. 4, but I can't find any statements referring to it in the main text. Why not include the following statements not only in the response to review but also into the manuscript?

"As can be seen in this new figure Fig. 4, D. ananassae was not more or less aggressive overall than other species, such as D. melanogaster, D. simulans, and D. atripex. Furthermore, the level and direction of aggression (heterospecific vs. conspecific) depended on the opponent species—e.g., the level of conspecific aggression was higher when species were paired with more distantly related species for D. ananassae, D. melanogaster, and D. simulans. Thus, it does not appear that the species-pair interactions that include D. ananassae are what solely drive the main patterns of our results. It will be important in future work to investigate broader sets of species, and in all combinations of species-pair interactions, in order to evaluate these trends in aggression behavior more broadly. It will be important in future work to investigate broader sets of species, and in all combinations of species-pair interactions, in order to evaluate these trends in aggression behavior more broadly."

We thank the reviewer for this comment and suggestion. In response to the first question regarding our reference to Figure 4 in the main text, this figure reference was included on Lines 215-217 in the previous version of the manuscript and is now on lines 205-206 in the revised manuscript. In response to the suggestion to include text that addresses the reviewer's concern about differential levels of aggression among the species, we have included a revised version of the quoted text from above in the revised version of the manuscript. This quoted text was taken from our previous response to review in the last submission, and we have revised it slightly to fit into the main text of the revised manuscript on lines 206-209.

2) I enjoyed the supplemental videos, and noticed that D. sim males were rigorously courting D. mel males in the Video S2. I suspect it induced more heterospecific aggression from D. mel males towards D. sim males. Because it suggests a possibility that non-aggressive behavior affects the results, I hope to see some explanations in the main text. "Mis-directed courtship

toward heterospecific males" could be a kind of reproductive interference if it happens under the natural condition.

We thank the reviewer for this comment and have included additional text in the Discussion (lines 308-318) to address this concern.

Reviewer: 2

Comments to the Author(s)

All my concerns have been addressed. The paper is clearer, and the figures are much improved. This study is a valuable contribution to our understanding of heterospecific aggression. I do think it's curious that there seems to be some interactive or condition-dependent effect of the species pair and conspecific aggression. I'm not sure what can be said about this, but it's interesting.

We thank the reviewer for this comment and agree that conspecific aggression did depend on the species-pair being assayed. This is one of the main results of our study, and it correlates with genetic distance between the species, as shown in Figure 5.